# Natural Compounds Isolated from *Stachybotrys chartarum* Are Potent Inhibitors of Human Protein Kinase CK2

**DOI:** 10.3390/molecules26154453

**Published:** 2021-07-23

**Authors:** Samer Haidar, Franziska M. Jürgens, Dagmar Aichele, Annika Jagels, Hans-Ulrich Humpf, Joachim Jose

**Affiliations:** 1Institut für Pharmazeutische und Medizinische Chemie, PharmaCampus, Westfälische Wilhelms-Universität Münster, Corrensstr. 48, 48149 Münster, Germany; shaid_01@uni-muenster.de (S.H.); franziska.juergens@uni-muenster.de (F.M.J.); dagmar.aichele@uni-muenster.de (D.A.); 2Faculty of Pharmacy, Damascus University, 17 April Street, Damascus P.O. Box 9411, Syria; 3Whitney Laboratory for Marine Bioscience, Department of Chemistry, University of Florida, Gainesville, FL 32080, USA; annika.jagels@whitney.ufl.edu; 4Institut für Lebensmittelchemie, Westfälische Wilhelms-Universität Münster, Corrensstr. 45, 48149 Münster, Germany; humpf@uni-muenster.de

**Keywords:** CK2, natural compounds, antiproliferation, *Stachybotrys*

## Abstract

A large number of secondary metabolites have been isolated from the filamentous fungus *Stachybotrys chartarum* and have been described before. Fourteen of these natural compounds were evaluated in vitro in the present study for their inhibitory activity towards the cancer target CK2. Among these compounds, stachybotrychromene C, stachybotrydial acetate and acetoxystachybotrydial acetate turned out to be potent inhibitors with IC_50_ values of 0.32 µM, 0.69 µM and 1.86 µM, respectively. The effects of these three compounds on cell proliferation, growth and viability of MCF7 cells, representing human breast adenocarcinoma as well as A427 (human lung carcinoma) and A431 (human epidermoid carcinoma) cells, were tested using EdU assay, IncuCyte^®^ live-cell imaging and MTT assay. The most active compound in inhibiting MCF7 cell proliferation was acetoxystachybotrydial acetate with an EC_50_ value of 0.39 µM. In addition, acetoxystachybotrydial acetate turned out to inhibit the growth of all three cell lines completely at a concentration of 1 µM. In contrast, cell viability was impaired only moderately, to 37%, 14% and 23% in MCF7, A427 and A431 cells, respectively.

## 1. Introduction

Protein kinase CK2, formerly known as Casein Kinase II, is an enzyme localized in both the cytoplasm and the nucleus of cells and appears to be constitutively active [1]. This enzyme is a Ser/Thr kinase, since most if not all of its substrates are phosphorylated at serine or threonine residues [2,3]. CK2 is involved in many cellular processes such as cell cycle regulation, gene expression, tRNA and rRNA synthesis, protein synthesis and degradation, cell growth and differentiation, embryogenesis and apoptosis [4]. CK2 enhances cancer progression by suppressing apoptosis and stimulating cell growth and hence is overexpressed in all types of tumors tested so far. Reducing CK2 activity to levels observed in non-neoplastic cells results in apoptosis, leading to tumor cell death [5]. CK2 is now considered an important drug target for cancer therapy [6]. Natural compounds are one of the major sources to discover new scaffolds for therapeutic purposes. An overwhelming number of natural products with diverse and complex structures which interact with a variety of biological targets are known [7]. Chemical scaffolds evolved by nature can be optimized in their biological activity with the aim of developing new drugs [7,8]. The identification of biological active candidates among natural compounds is a challenge since natural products are mainly found in complex mixtures and rather small amounts. Recently, Newman et al. reviewed the use of natural products as a source of new drugs over a period of three decades and noticed that around half of the small molecules approved as drugs are natural or naturally derived compounds. Almost two thirds of cancer chemotherapeutics developed from 1981 to 2010 were derived from natural products [8]. This reflects the importance of natural or natural-product-derived compounds as an important source for drug discovery. 

Different types of inhibitors of protein kinase CK2 were developed during the last decade [6,9]. Among them, several natural compounds were identified, such as emodin, elagic acid and others [10]. Recently, using an in silico screening approach, the natural compound bikaverin was identified as a potent inhibitor of CK2 [11,12]. Most of the ATP competitive inhibitors of CK2 share common structural features; they consist of planar scaffolds mimicking the adenine or the purine of the ATP with different substituents. Compounds that have been isolated in a previous study from fungal cultures of different *Stachybotrys* strains share some intriguing structural similarities to known CK2 inhibitors. The natural compounds isolated from the genus *Stachybotrys* belong to different structural groups, including meroterpenoids containing a chromene ring moiety as well as macrocyclic trichothecenes and phenylspirodrimanes [13,14]. Two meroterpenoids exhibited moderate cytotoxic effects on human liver carcinoma cells (HepG2) [14]. In the present follow-up study, fourteen compounds isolated from *Stachybotrys chartarum* were tested in silico and in vitro on inhibition of CK2. The most active natural compounds were evaluated in terms of impairing cell viability and suppressing cell proliferation in different tumor cell lines.

## 2. Results

### 2.1. Inhibition of Human Protein Kinase CK2 by Natural Compounds from Stachybotrys chartarum

Fourteen compounds which were isolated from *Stachybotrys chartarum* as described earlier [13,14] were tested for inhibition of human protein CK2 by a CE-based activity assay [15,16]. These compounds belong to three different scaffolds (Figure 1), namely: stachybotrychromenes A–C, phenylspirodrimanes (stachybotrydial, stachybotrydial acetate, acetoxystachybotrydial acetate, stachybotrysin B, stachybotrysin C, stachybotrylactam acetate, L-671, stachybonoid D, stachybotryamide) and macrocyclic trichothecenes (satratoxin G, H). All compounds were initially tested for inhibition of CK2 at a concentration of 10 µM. For compounds showing an inhibition of 50% or more in comparison to the control with DMSO, an IC_50_ value was determined (Table 1).

Five compounds, stachybotrychromene C, stachybotrydial acetate, acetoxystachybotrydial acetate, stachybotrydial and stachybotrysin B, inhibited human CK2 to more than 50% at a concentration of 10 µM (Table 1). For these compounds, the IC_50_ values were determined using nine different concentrations ranging from 0.001 µM to 100 µM in appropriate intervals. The most active compound appeared to be stachybotrychromene C with an IC_50_ value of 0.32 µM. Stachybotrychromenes A and B were less active, which might reflect the importance of the two aldehyde groups on the phenol ring, obviously increasing inhibition substantially. All of the phenylspirodrimanes showed inhibition to some extent as well. The most active inhibitors from this group turned out to be stachybotrydial acetate followed by acetoxystachybotrydial acetate, with IC_50_ values of 0.69 µM and 1.86 µM, respectively. Stachybotrydial and stachybotrysin B were obviously less active inhibitors of CK2, with IC_50_ values of 4.43 µM and 13.42 µM, respectively. This suggests that the two aldehyde groups on the phenol ring together with the one acetoxy group on the drimane skeleton are important for the inhibitory effect. On the other hand, the two macrocyclic trichothecenes, as tested, showed almost no inhibition (Table 1). For comparison, the inhibitory activity of emodin, which is also a natural compound with known CK2 inhibition, was tested in the same assay and an IC_50_ value of 0.60 µM was determined, which was in agreement with the value described earlier [10].

### 2.2. Molecular Docking

The three most active compounds: stachybotrychromene C, stachybotrydial acetate and acetoxystachybotrydial acetate were docked in the ATP binding site of the CK2 crystal structure (PDB ID: 3C13, resolution 1.95 Å) [17]. Prior to docking, a conformational search for all compounds was carried out by Molecular Operating Environment (MOE) and the resulting conformations were used for the docking study. The three compounds fit well in the ATP binding site of the enzyme, with an S score of −7.4925 (stachybotrychromene °C), −5.1901 (stachybotrydial acetate) and −5.0028 (acetoxystachybotrydial acetate), which are in the same range as the S score of co-crystallized emodin (−5.852), which has an IC_50_ value of 0.60 µM. Figure 2 presents the “Ligand-Protein Contacts Interactions” in 2D and 3D of the three compounds generated with MOE. It appeared that the compounds fit well in the ATP binding site and create adequate bonds. Stachybotrychromene C is able to create a π-hydrogen bond with the “gatekeeper” residue Ph113 [18], whereas stachybotrydial acetate and acetoxystachybotrydial acetate bind to Val53 by a π-hydrogen bond (Figure 2).

### 2.3. ATP Competitive Mode of Action

Molecular docking revealed that the three most active substances fit well in the ATP binding site. Therefore an ATP-competitive mode of inhibition was investigated. IC_50_ values of acetoxystachybotrydial acetate were determined in dependency of the co-substrate ATP. Acetoxystachybotrydial acetate was chosen for this purpose because it was available in sufficient quantity. Nine different concentrations of the compound ranging from 0.001 µM to 100 μM were tested for CK2 inhibition in the presence of three different ATP concentrations (10 µM, 50 µM, 100 µM). The IC_50_ values were observed to increase linearly with the ATP concentrations, clearly indicating an ATP competitive mode of inhibition (Figure 3). In the resulting graph, the K_i_ value was determined by the interception of the regression line with the Y-axis, and it turned out to be 1.41 μM (Figure 3).

A reaction of aldehyde groups appearing at position 4´and 5´ of the phenylspirodrimanes such as acetoxystachybotrydial acetate (Figure 1) with amino acids in proteins has been reported before [19,20]. The ATP dependency of the IC_50_ value as shown above is a clear indicator that the interaction between the compound and CK2 seems to be specific and is not due to any non-specific reaction of the dialdehydes of phenylspirodrimanes with amines in a protein. Finally, to rule out this aspect, the inhibitory effect of acetoxystachybotrydial acetate on lactate dehydrogenase (LDH) was determined. Even at a concentration of 10 µM, no inhibition of LDH by acetoxystachybotrydial acetate was observed, indicating no non-specific interaction of acetoxystachybotrydial acetate by coupling to protein amines (data not shown).

### 2.4. Inhibition of Tumor Cell Proliferation

In order to evaluate a potential anticancer effect of the three most active compounds, MCF7-NucLight Green cells (human breast adenocarcinoma cell line), A427 (human lung carcinoma cell line) and A431 (human epidermoid carcinoma cell line) cells were treated with different concentrations of stachybotrychromene C, stachybotrydial acetate and acetoxystachybotrydial acetate. First, the inhibition of tumor cell growth was evaluated by determining the confluence after treatment with the selected compounds using an IncuCyte^®^ live cell imager. Cell growth of MCF7 NucLight Green and A431 cells was completely blocked by the addition of 1 µM of acetoxystachybotrydial acetate (Figure 4), whereas the growth of A427 was strongly impaired at this concentration. Stachybotrydial acetate showed no effect on cell growth of MCF7 NucLight green or A427 and A431 cells in a concentration of 1 µM, but growth of all cell lines was completely blocked at a concentration of 100 µM in the case of this compound. Stachybotrychromene C showed no inhibition of growth in any cell line at a concentration of 1 µM. However, it was able to reduce the growth of MCF7 NucLight Green cells, at a concentration of 100 µM (data not shown), whereas no other cell line was affected.

In Figure 5, the morphology of MCF7 NucLight Green after treatment with 1 µM of acetoxystachybotrydial acetate for 24 h and 48 h is shown. Cells treated with 1% DMSO were used as a control and the morphology was recorded by the IncuCyte^®^ live cell imager. Compared with the control, the total number of MCF7 NucLight Green cells was strongly reduced after incubation with 1 µM of acetoxystachybotrydial acetate for 48 h. A similar image could be observed after 24 h as well. Furthermore, shrinkage of cells was monitored and their morphology changed to a more spherical shape after treatment with the compound. A427 and A431 cells exhibited similar changes in morphology after treatment with 1 µM of acetoxystachybotrydial acetate for 48 h (Figure 5).

Inhibition of cell proliferation was determined by the EdU assay in MCF7 NucLight Green cells after 24 h incubation with 1 µM or 100 µM acetoxystachybotrydial acetate. Proliferating cells are recognized by red fluorescence due to coupling of 5-TAMRA-PEG3-azide to the modified nucleotide incorporated in case of DNA replication (Figure 6).

Less than 1% of MCF7 cells were still proliferating after treatment with 1 µM of acetoxystachybotrydial acetate (Figure 7). To obtain a similar effect on MCF7 cells with stachybotrydial acetate, a concentration of 100 µM was necessary, but still, the number of proliferating cells was obviously higher than with 1 µM acetoxystachybotrydial acetate. In contrast, stachybotrychromene C showed only negligible effects on MCF7 cell proliferation, even at concentrations of 100 µM. To evaluate the results obtained with acetoxystachybotrydial acetate, emodin, a known natural inhibitor of CK2 as described above, was applied in the same assay. At a concentration of 100 µM emodin, 5% of the MCF7 cells were proliferating, at a concentration of 10 µM, 77% were proliferating and at a concentration of 1 µM, no effect on MCF7 tumor cell proliferation was detectable (data not shown). This indicates that acetoxystachybotrydial acetate is a more potent inhibitor of tumor cell proliferation, at least in the case of MCF7 breast cancer cells, than the known inhibitor of human protein kinase CK2, emodin.

The EC_50_ value of acetoxystachybotrydial acetate for the inhibition of tumor cell proliferation after 24 h was determined for MCF7 NucLight green cells (Figure 8) and appeared to be 0.39 µM.

Furthermore, cell viability was tested with the MTT assay for the three cell lines mentioned above. Stachybotrychromene C, stachybotrydial acetate and acetoxystachybotrydial acetate were applied in concentrations of 1 µM and 100 µM for 48 h (Figure 9). Treatment of A427 and A431 cells with 1 µM of stachybotrychromene C for 48 h caused a moderate reduction of cell viability by 32% and 27%, respectively. Almost no further reduction was seen with a concentration of 10 µM (data not shown) and only a slight increase of inhibitory effect by nearly 10% was obtained with a concentration of 100 µM (Figure 9). Viability of MCF7 cells was not decreased by adding a concentration of 100 µM, but was increased to 155% of the control by adding 1 µM of stachybotrychromene C. The same was observed for MCF7 cells when incubated with 1 µM of stachybotrydial acetate. At a concentration of 100 µM, stachybotrydial acetate reduced cell viability by 86%, 82% and 93% of the control in MCF7, A427 and A431, respectively, resulting in only residual cellular activity. At a concentration of 10 µM, reduction of cell viability in A427 and A431 was not significant, whereas cell viability of MCF7 was slightly increased (by 20%, data not shown). Treatment with 1 µM of the compound results in a reduction of cell viability in A427 and A431 cells to 86% and 73% of the control. The most striking effect on cell viability was observed for acetoxystachybotrydial acetate with residual 2% (MCF7), 12% (A427) and 5% (A431) of cell viability after 48 h at 100 µM. At a concentration of 1 µM, a reduction of cell viability to 37% for MCF7, 12% for A427 and 23% for A431 cells was observed.

## 3. Discussion

Protein kinase CK2, which is involved in cell proliferation and survival, is overexpressed in different human cancers. Selecting new hits with significant inhibitory activity toward this enzyme will definitely contribute to finding new drugs for the treatment of cancers from different origins. In this work, we demonstrated that three isolated natural products from *Stachybotrys chartarum* [13,14], namely, stachybotrychromene C, stachybotrydial acetate and acetoxystachybotrydial acetate were potent CK2 inhibitors with IC_50_ values in the low µM range. Stachybotrydial acetate and stachybotrychromene C showed only minor effects on tumor cell growth and tumor cell proliferation even at 100 µM concentration, whereas acetoxystachybotrydial acetate almost completely blocked tumor cell growth and tumor cell proliferation at a concentration of 1 µM. For the inhibition of tumor cell proliferation in MCF7 breast cancer cells, an EC_50_ value of 0.39 µM was obtained. In contrast to this, effects on cell viability were not as strong as effects on cell growth and proliferation. This implies that the anticancer effects are not simply due to cytotoxic effects. It is noteworthy that the EC_50_ value of acetoxystachybotrydial acetate for MCF7 cells (0.39 µM) was in the submicromolar range, and even much lower compared to the EC_50_ values of other human CK2 inhibitors such as bikaverin (1.97 µM) [12] or DMAT-derived hydroxamates (9.02; 9.97 and 15.66 µM) [21]. As described, acetoxystachybotrydial acetate was very effective in different cancer cell lines, whereas the most active compound towards the target enzyme CK2, which was stachybotrychromene C, showed much less effect in the cellular assays. The possibility cannot be excluded that this is partially due to the logP values of the compounds, which were calculated to be 6.04 for stachybotrychromene C, 2.81 for acetoxystachybotrydial acetate and 3.43 for stachybotrydial acetate. Overly high logP values, as seen here for stachybotrychromene C, could have a negative effect on membrane permeability of a compound and hence on its cellular uptake.

Cytotoxicity of stachybotrychromenes A–C was determined earlier on HepG2 cells, using a resazurin reduction assay and no or moderate effects were detectable [14]. These results are in accordance with the results obtained in this study for stachybotrychromene C with three different cancer cell lines, MCF7, A427 and A431. It is important to note that several meroterpenoid compounds were reported to have broad biological activities, such as immune toxicity, neurotoxicity, cytotoxicity, fibrinolysis, antiviral and anti-plasmodial activity [22,23,24]. However, to the best of our knowledge, none of these compounds were evaluated towards inhibition of any protein kinase before. In this work, several compounds isolated from the filamentous fungus *Stachybotrys chartarum* were shown to be potent CK2 inhibitors; among them, acetoxystachybotrydial acetate showed considerable effects on cell viability and cell proliferation. The compounds described in this work as CK2 inhibitors belong to two different backbones and can serve as a starting point for the discovery of more active derivatives.

## 4. Materials and Methods

### 4.1. The Compounds

Emodin was purchased from Merck Millipore (Darmstadt, Germany). All commercial reagents were of the highest available purity grade. The compounds were dissolved in dimethyl sulfoxide (DMSO) and the stock solutions were stored at −20 °C and warmed to 25 °C just before use. The isolation of compounds obtained from *Stachybotrys chartarum* has been described earlier [13,14].

### 4.2. Inhibition of Human CK2 Holoenzyme

The selected natural compounds were tested for their inhibitory activity towards the human recombinant CK2 holoenzyme in a Capillary Electrophoresis (CE)-based assay following the procedure described earlier [15]. The synthetic peptide RRRDDDSDDD was used as a substrate, which was reported to be efficiently phosphorylated by CK2. The purity of the CK2 holoenzyme as applied was beyond 99%. For initial testing, inhibition was determined relative to the controls at inhibitor concentrations of 10 μM in DMSO as a solvent. The reaction with pure solvent without inhibitor was used as a control and set to 0% inhibition. Reactions without CK2 were used as negative control and were taken as 100% inhibition. IC_50_ values were determined by measuring CK2 inhibition at nine different concentrations of inhibitors ranging from 0.001 µM to 100 µM in appropriate intervals and calculated from the resulting dose-response curve [16]; Prism 6 (GraphPad Software, San Diego, CA, USA) was used to estimate the IC_50_ values. For the determination of the mode of inhibition, the ATP concentration in the assay buffer was varied to 10, 50 and 100 μM, while the rest of the procedure was identical to the IC_50_ determination described above.

### 4.3. Cultivation of Cancer Cell Lines

MCF7 NucLight Green cells (human breast adenocarcinoma cell line), provided from Essen Bioscience, Royston, UK, were cultured in Eagle’s Minimum Essential Medium (EMEM) supplemented with 2 mM L-glutamine, 0.01 mg/mL human recombinant insulin, 10% Fetal Calf Serum (FCS), 1% Penicillin/Streptomycin (Pen/Strep) and 0.5 µg/mL puromycin (every second passage) [25]. Human epidermoid carcinoma cells A431 (kindly provided by the Department of Experimental Tumor Biology, WWU Münster, Germany) were cultured in Dulbecco’s Modified Eagle’s Medium (DMEM) high glucose, supplemented with 2 mM L-glutamine and 10% FCS. For cultivation of A427 human lung carcinoma cell line (purchased from German Collection of Microorganisms and Cell Cultures (DSMZ), No. ACC234, Braunschweig, Germany), Roswell Park Memorial Institute, medium (RPMI1640) supplemented with 2 mM L-glutamine and 10% FCS was used. For testing CK2 inhibitors on cell growth, cell proliferation or cell viability, cells were seeded at a density of 5 × 10^3^ cells per well (A431, A427) or 1 × 10^4^ cells per well (MCF7 NucLight Green) into 96-well culture plates. After overnight incubation at 37 °C in a humidified atmosphere (5% CO_2_), seeding medium was removed and replaced with fresh medium containing the compounds at 1, 10 or 100 μM. DMSO at a final concentration of 1% served as a control. Cells were subsequently incubated for a further 24 or 48 h at 37 °C in a humidified atmosphere (5% CO_2_). 

### 4.4. Cell Proliferation

Cell proliferation of MCF7 NucLight green cells was quantified by the EdU-click assay (Baseclick BCK-EdU555-1, Baseclick GmbH, Munich, Germany). The nucleoside analog 5-ethynyl-2′-deoxyuridine (EdU) is incorporated during DNA synthesis, and a 5-carboxytetramethylrhodamin-triethylene glycol-azide (5-TAMRA-PEG3-azide) fluorophore used for detection is coupled by click reaction. MCF7 NucLight green cells were incubated with the inhibitors for 24 h. Medium was replaced with medium containing 10 µM EdU. After a further 24 h, the 5-TAMRA-PEG3-azide fluorophore was added as described in the Baseclick assay manual. The results were given as a percent ratio of proliferating cells relative to cells treated with 1% DMSO as a control. The effects of CK2 inhibitors were analyzed in triplicate and all experiments were repeated three times independently.

### 4.5. Cell Viability Assay 

The effect of CK2 inhibitors on the viability of MCF7 NucLight green, A427 and A431 cells was evaluated using MTT assay [26]. This is a colorimetric assay, which measures the conversion of 3-(4,5-dimethylthiazol-2-yl)-2,5-diphenyltetrazolium bromide (MTT) into violet formazan that is produced by succinate dehydrogenase of the intact mitochondria in viable cells. MTT assay was performed in 96-well plates. Cells were seeded at a density of 5 × 10^3^ cells per well (A431, A427), or 1 × 10^4^ cells per well (MCF7 NucLight Green). After overnight incubation, seeding medium was removed and replaced with fresh medium containing the inhibitor at 1, 10 or 100 μM. DMSO at a final concentration of 1% served as a control. Cells were incubated for 48 h at 37 °C in a humidified atmosphere (5% CO_2_). Afterwards, MTT reagent (Sigma Aldrich, Steinheim am Albuch, Germany) was added at a final concentration of 0.5 mg/mL. After incubation for 2 h at 37 °C, the medium was discarded and 200 µL DMSO was added to solve the resulting formazan. After mixing, the absorption was determined at 570 nm with a reference wavelength of 630 nm using a microplate reader. CK2 inhibitors were assayed in triplicate, and the experiments were repeated three times.

### 4.6. IncuCyte^®^ Cytotoxicity Assay

The IncuCyte^®^ S3 Live-Cell Analysis System (Ann Arbor, MI, USA) was used for monitoring cell growth and to determine the cytotoxic effect of tested compounds in the three cell lines. This technique allows an automated in-incubator method of live cell monitoring. The concentration-dependent growth-enhancing/growth-inhibitory activity of the tested compound was evaluated against the above mentioned cell lines. After addition of inhibitors in concentrations of 1 µM, 10 µM and 100 µM, cell growth was monitored for 48 h by generating phase contrast images. Growth curves were generated by the algorithm in the “2017A Rev2” software from data points acquired during 2 h interval imaging. All samples were plated in triplicate.

### 4.7. LDH Assay

For the determination of enzymatic activity of lactate dehydrogenase (LDH), 0.25 µg of LDH-5 from rabbit muscle (Roche Diagnostics GmbH, Mannheim, Germany) was incubated in phosphate buffered saline (PBS) containing 0.25 mM nicotinamide adenine dinucleotide (NADH) and 0.625 mM pyruvate in a total volume of 1 mL at room temperature. Absorption at 340 nm was monitored every 5 s within 5 min. For the determination of inhibitory activity of phenylspirodrimanes towards LDH, acetoxystachybotrydial acetate was added in a final concentration of 10 µM; 0.1% DMSO served as a control.

### 4.8. Computational Study

Molecular Operating Environment software package (MOE, Chemical Computing Group, Montreal, QC, Canada) [27] running on Intel Core, i5-6500CPU, 3.20 GHz processor was used to perform this study.

#### 4.8.1. Database Generation

The selected compounds were rebuilt with MOE building option implemented in the software. The compounds were optimized by adding hydrogen atoms using the option of MOE software. The energy of the compounds was minimized using the following parameters: gradient: 0.05, Force Field: MMFF94X, Chiral constraint and Current Geometry. The conformation methodology was used to develop low energy conformations for each compound, applying the LowModMD method with RMS gradient of 0.05; all other parameters were used as default. All of the compounds and their conformations were saved in mdb database and employed for docking studies later.

#### 4.8.2. Protein Structure Preparation and Molecular Docking

Three-dimensional structure of the CK2 complex with emodin was obtained from the Protein Data Bank (PDB) using PDB ID: (3C13) having a resolution of 1.95 Å [17]. The structure was optimized by using QuickPrep function implemented in the MOE software package [27]. In the second step, energy minimization was performed using default parameters, where the force field was Amber 10. The docking of the selected compounds into the active site of the CK2 enzyme (3C13) was achieved using MOE-Dock implemented in MOE. The docking parameters were set as Rescoring 1: London dG, Placement: triangle matcher, Retain 30, Refinement Force field and Rescoring 2: GBVI/WSA dG, Retain 30. The docking simulations of MOE predicted the most favorable conformations of each compound to obtain the minimum energy structure. The top conformation for each compound was selected based on the S score and visual inspection in 2D and 3D. Prior to docking, the initial ligand from the complex structure was extracted. For the scoring function, lower scores indicated more favorable poses. The unit for the scoring function was Kcal/mol, and the S score referred to the final score, which was the score of the last stage that was not set to None. The Lig X function in MOE was used for conducting interactive ligand modification and energy minimization in the active site of the receptor. 

### 4.9. Statistical Analysis

All statistical analyses and statistical diagrams were generated with Prism 6 software (GraphPad Software, San Diego, CA, USA).

## Figures and Tables

**Figure 1 molecules-26-04453-f001:**
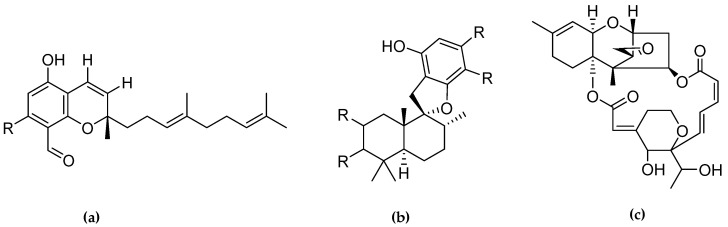
Chemical backbones of the tested compounds: (**a**) stachybotrychromenes, (**b**) phenylspirodrimanes and (**c**) macrocyclic trichothecenes.

**Figure 2 molecules-26-04453-f002:**
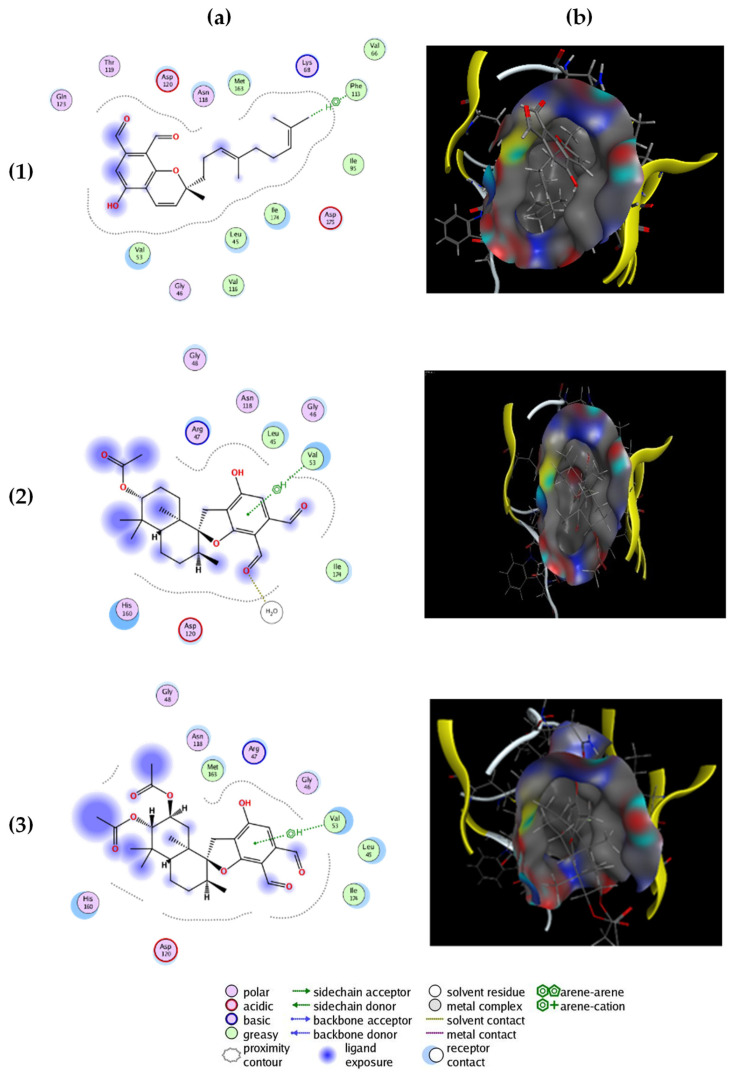
(**a**) Two-dimensional interaction between the three best compounds and the ATP binding site and (**b**) snapshots representing the three-dimensional docking complex of the selected compounds with the binding site of CK2: (1) stachybotrychromene C, (2) stachybotrydial acetate and (3) acetoxystachybotrydial acetate.

**Figure 3 molecules-26-04453-f003:**
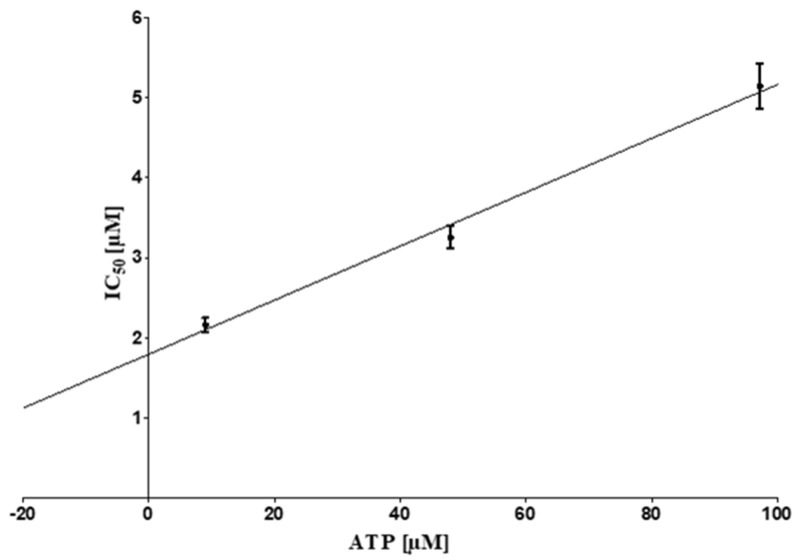
ATP-competitive inhibition of human CK2 by acetoxystachybotrydial acetate. IC_50_ values with three different ATP concentrations were determined using nine different concentrations of the inhibitor ranging from 0.001 µM to 100 μM and plotted against the corresponding ATP concentrations. Each IC_50_ value was determined three times independently. Mean values with standard deviation are given. The Ki value is defined as the Y-axis intercept and was determined to be 1.41 μM (R^2^ = 0.97).

**Figure 4 molecules-26-04453-f004:**
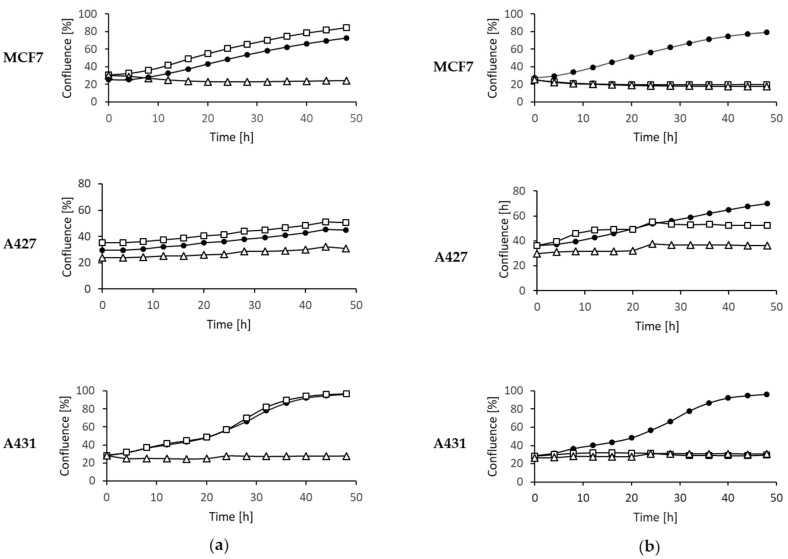
Inhibition of tumor cell growth by (**a**) stachybotrydial acetate and (**b**) acetoxystachybotrydial acetate. MCF7 NucLight Green cells, A431 cells and A427 cells were treated with 1 µM (□) and 100 µM (Δ) of stachybotrydial acetate or acetoxystachybotrydial acetate or 1% DMSO as control (●) for 48 h. Cell confluence was monitored over 48 h using IncuCyte^®^ S3 live cell imaging system and analyzed using the IncuCyte^®^ S3 2017A software.

**Figure 5 molecules-26-04453-f005:**
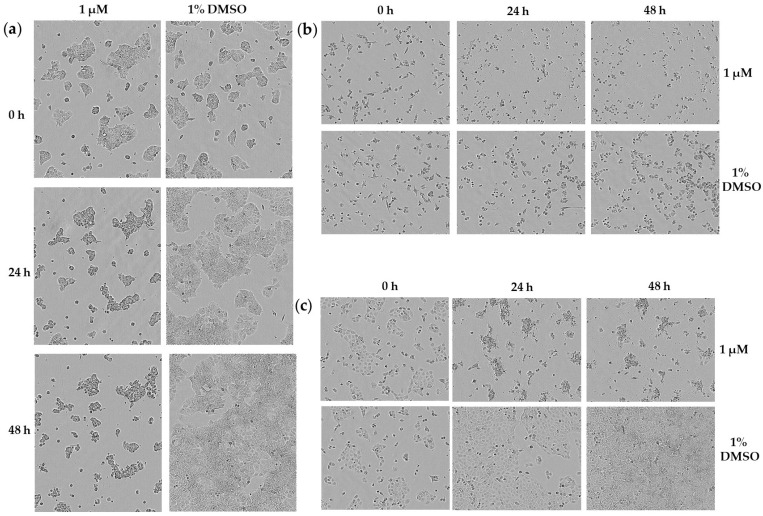
Phase contrast images of (**a**) MCF7 cells, (**b**) A427 cells and (**c**) A431 cells treated with acetoxystachybotrydial acetate in a concentration of 1 µM and 1% DMSO as control at 0 h, 24 h and 48 h after treatment. Images were taken at tenfold magnification on IncuCyte^®^ live cell imager.

**Figure 6 molecules-26-04453-f006:**
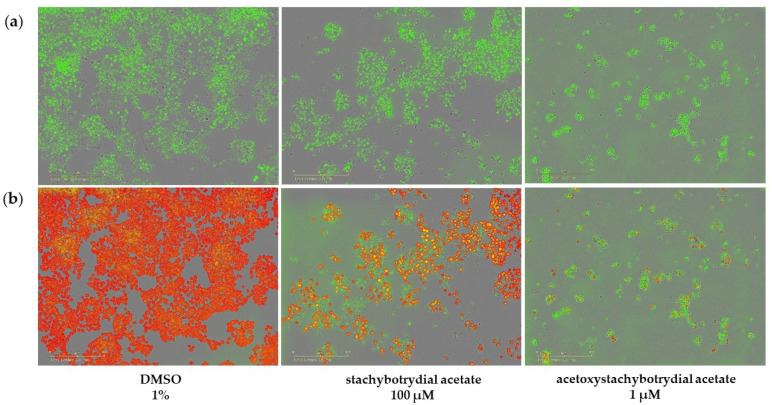
Fluorescence images of MCF7 NucLight green cells treated with 1% DMSO, 100 µM stachybotrydial acetate or 1 µM acetoxystachybotrydial acetate for 24 h. (**a**) Detection of cells by their green fluorescent nuclei. (**b**) Proliferating cells were detected by an additional staining of cell nuclei by the EdU assay using 5-TAMRA-PEG3-azide as a coupled fluorophore. Thus, proliferating cells were monitored by red fluorescence. The pictures in lane (**b**) are an overlay of the green fluorescence images of MCF7 NucLight green and the red fluorescence images showing TAMRA-labeled proliferating cells. Cells that are emitting only green fluorescence are not proliferating, in contrast to those emitting an additional red fluorescence. Images were taken at tenfold magnification on IncuCyte^®^ live cell imager.

**Figure 7 molecules-26-04453-f007:**
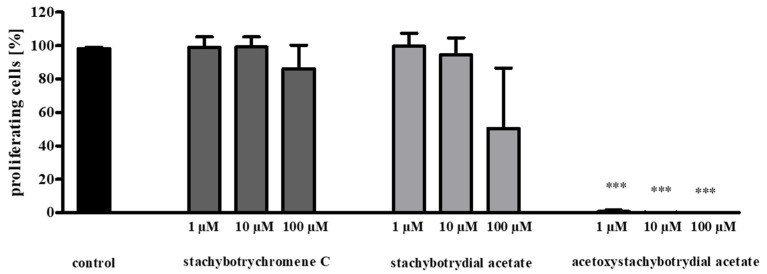
Quantification of the antiproliferative effect of stachybotrychromene C, stachybotrydial acetate and acetoxystachybotrydial acetate on MCF7 NucLight Green cells after 24 h of incubation. Numbers of proliferating cells were determined by EdU assay. Results are shown as a percent of proliferating cells relative to control cells (with 1% DMSO) and represent the mean (± SD) of three independent experiments. *** *p* < 0.001.

**Figure 8 molecules-26-04453-f008:**
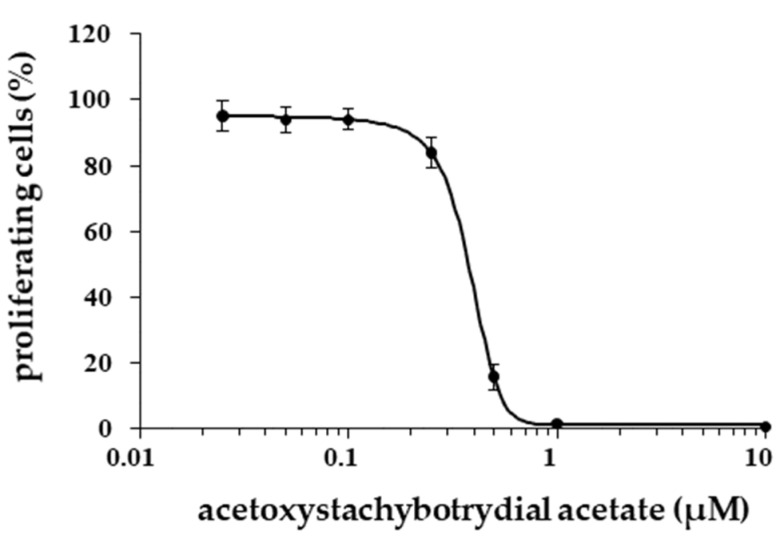
Dose-dependent inhibition of MCF7 NucLight green cell proliferation by acetoxystachybotrydial acetate using EdU assay. The EC_50_ is 0.39 ± 0.02 µM, which was determined in three independent replications. Mean values with standard deviations are given.

**Figure 9 molecules-26-04453-f009:**
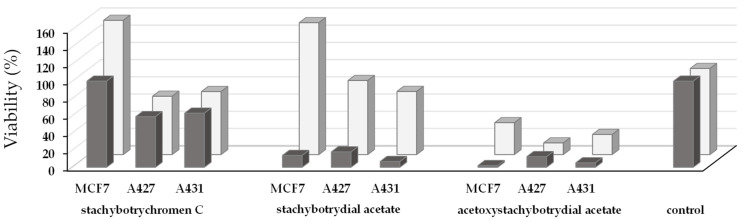
Cell viability of MCF7, A427 and A431 cells after 48 h treatment with stachybotrychromene C, stachybotrydial acetate and acetoxystachybotrydial acetate as tested by MTT assay at a concentration of 100 µM (dark gray) and at a concentration of 1 µM (light gray) in three independent experiments.

**Table 1 molecules-26-04453-t001:** Inhibition of human protein kinase CK2 by 14 natural compounds isolated from *Stachybotrys chartarum*.

Compound Name	Chemical Structure	% Inhibition at 10 µM (IC_50_ µM) ^a^
Stachybotrychromene A	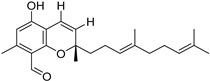	26 (n.d.)
Stachybotrychromene B	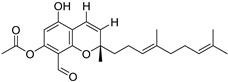	47 (n.d.)
Stachybotrychromene C	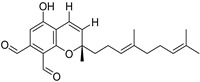	95 (0.32 ± 0.20)
Stachybotrydial acetate	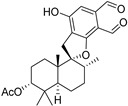	96 (0.69 ± 0.15)
Acetoxystachybotrydial acetate	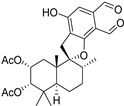	87 (1.86 ± 0.36)
Stachybotrydial	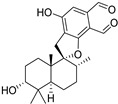	72 (4.43 ± 0.30)
Stachybotrysin B	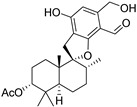	50 (13.42 ± 2.40)
Stachybotrylactam acetate	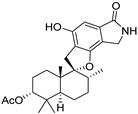	44 (n.d.)
L-671	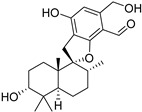	42 (n.d.)
Stachybonoid D	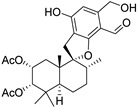	47 (n.d.)
Stachybotrysin C	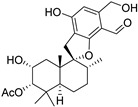	37 (n.d.)
Stachybotryamide	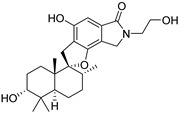	21 (n.d.)
Satratoxin G	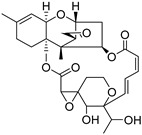	12 (n.d.)
Satratoxin H	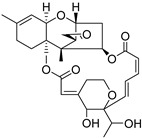	10 (n.d.)

^a^ Mean values ± standard deviation (SD) from three independent experiments. n.d.: not determined.

## Data Availability

Data reported in this study is contained within the article. The underlying rawdata is available on request from the corresponding author.

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
