# Peer review of "Natural Compounds Isolated from *Stachybotrys chartarum* Are Potent Inhibitors of Human Protein Kinase CK2"

_molecules, 2021, doi:10.3390/molecules26154453_

Round 1

Reviewer 1 Report

The paper entitled " Natural Compounds isolated from Stachybotrys chartarum are potent inhibitors of human protein kinase CK2“ submitted by Joachim Jose et al. provides very useful data for the chemistry community; in particular for scientists in the area of medicinal chemistry and it also shows new structures with challenging frameworks for natural product chemists.

The data presented are solid and the biological activities obtained are very useful starting points for medicinal chemistry programs. Finally, the target is of great interest for the pharmaceutical industry. In summary, the paper should be published in molecules.

Author Response

We are grateful to the reviewer for this evaluation.

Reviewer 2 Report

Dear Authors, This manuscript described 14 known compounds found in Author's previous work were intensively evaluated for their inhibitory activity against human protein kinase CK2 for the first time, and cytotoxicity against human cancer cell lines MCF7, A427 and A431. As a result, stachybotrychromene C and stachybotrydial acetate showed promising activity as CK2 inhibitor. 1) The identified 14 known compounds should be cited. 2) The abstract might have to re-phase it, since this work is a follow-up of Author's previous work [ref 13]. Author's previous work [ref 13] have reported all the compounds described within this manuscript, which should be mentioned in the abstract and introduction. This is a follow-up work using previous reported compounds for human protein kinase CK2 inhibitory and human cancer cell lines cytotoxic assay.

Author Response

Dear Authors, This manuscript described 14 known compounds found in Author's previous work were intensively evaluated for their inhibitory activity against human protein kinase CK2 for the first time, and cytotoxicity against human cancer cell lines MCF7, A427 and A431. As a result, stachybotrychromene C and stachybotrydial acetate showed promising activity as CK2 inhibitor. 1) The identified 14 known compounds should be cited. 2) The abstract might have to re-phase it, since this work is a follow-up of Author's previous work [ref 13]. Author's previous work [ref 13] have reported all the compounds described within this manuscript, which should be mentioned in the abstract and introduction. This is a follow-up work using previous reported compounds for human protein kinase CK2 inhibitory and human cancer cell lines cytotoxic assay.

We fully agree and changed the abstract and the introduction as follows (and as can be seen in the track change mode of the manuscript):

Abstract:

A large number of secondary metabolites have been isolated from the filamentous fungus Stachybotrys chartarum and described before. Fourteen of these natural compounds were evaluated in vitro in the present study on their inhibitory activity....

Introdution:

... Compounds that have been isolated in a previous study from fungal cultures of different Stachybotrys strains share some intriguing structural similarities to known CK2 inhibitors [13,14]. ...

In the present follow up study, fourteen compounds isolated from Stachybotrys chartarum ...

In addition, we corrected two typhos in the address field andadded a higher resolusion figuure 5.
